# Comparison Study between Artificial Urinary Sphincter and Adjustable Male Sling: A Propensity-Score-Matched Analysis

**DOI:** 10.3390/jcm12175489

**Published:** 2023-08-24

**Authors:** Paolo Geretto, Enrico Ammirati, Marco Falcone, Alberto Manassero, Marco Agnello, Marcello Della Corte, Paolo Gontero, Alessandro Giammò

**Affiliations:** 1Division of Neuro-Urology, Department of Surgical Sciences, Città della Salute e della Scienza Hospital, University of Turin, 10126 Turin, Italyamanassero@cittadellasalute.to.it (A.M.);; 2Urology Clinic-A.O.U. “Città della Salute e della Scienza”-Molinette Hospital, University of Turin, 10100 Turin, Italy; pgontero@cittadellasalute.to.it; 3Division of Urology, Department of Oncology, San Luigi Gonzaga Hospital, University of Turin, Regione Gonzole 10, 10043 Orbassano, Italy

**Keywords:** artificial urinary sphincter, urinary incontinence, male sling, male urinary incontinence

## Abstract

Aims: This study aimed to compare the outcomes of the AUS and an adjustable male sling (ATOMS^TM^). Methods: It was a retrospective observational cohort study with two arms. Propensity score matching (PSM) was performed in order to limit selection bias and, consequently, a comparison between groups in terms of functional outcomes (24 h pad test and perception of improvement questionnaires), complications (overall complications, high-grade complications, reinterventions and explantations) and device survival was performed. Results: 49 patients in both arms were included. The baseline characteristics were similar between the groups. The mean follow up was 43 ± 35 months. Dryness was achieved in 22 patients (44.9%) in the AUS group and 11 (22.5%) in the sling group (*p* = 0.03). A total of 40 patients declared themselves well improved in the sling group (81%), while 35 (71%) declared the same in the AUS group (*p* = 0.78). The AUS was associated with more high-grade complications, reinterventions and explantations than the ATOMS^TM^. Survival at 60 months was 82 ± 9% in the sling group and 67 ± 7% in the AUS group (*p* = 0.03). Conclusions: While the AUS may be characterized by a higher dry rate, it has an increased risk of high-grade complications and reinterventions. It is proposed that the ATOMS prosthesis can be successfully used for patients who require a less invasive procedure that maintains good functional outcomes.

## 1. Introduction

Urinary incontinence (UI) is a relatively common complaint in male individuals who have undergone types of low urinary tract surgery, such as radical prostatectomy or prostatic disobstruction for benign prostatic hyperplasia (BPH). Indeed, up to 5–9% of patients subjected to radical prostatectomy report stress urinary incontinence (SUI) at 24 months [1]. Therefore, choosing appropriate treatments for post-surgical SUI represents an epidemiologically relevant clinical need and can greatly influence patient well-being.

Among all available anti-incontinence devices, the artificial urinary sphincter (AUS) was the first device introduced to the clinical scene, in 1973 [2]; its mechanism of action is a continuous circumferential compression of the bulbar urethra. The AMS 800^TM^ is today the most used AUS and consists of three components: a peri-urethral cuff which compresses the urethra, a reservoir and a scrotal activator. Over time, the AMS 800 has shown a good clinical effectiveness, having an acceptable safety profile with a dry rate exceeding 80% in some studies. Together with the AUS, new devices such as male slings and compression systems [3] have been developed over the years with the aim of overcoming the limitations of the AUS (e.g., high cost, difficulty to handle and high number of complications). Among these devices, male slings are currently largely used in clinical practice. Although fixed male slings (e.g., Advance^TM^) are the most used worldwide, adjustable male slings are progressively taking hold. The main advantage of adjustable male slings is their ability to be regulated after implantation in order to achieve better continence in case of persistent or recurrent SUI [4,5,6]. Among these slings, the Adjustable Trans Obturator Male System (ATOMS^TM^), a third-generation device, has shown promising clinical results in terms of safety and effectiveness, based on large cohorts of patients and long follow-up periods, with some authors suggesting it could serve as an alternative to the AUS in selected cases [7].

However, even if growing evidence for the clinical value of male slings is available, the AUS still constitutes the reference standard for the treatment of male SUI, as stated by the European Association of Urology (EAU) guidelines [8], because of the lack of high-quality comparison studies between the AUS and male slings. Indeed, only a single randomized controlled trial comparing the AUS to a fixed male sling in the treatment of male SUI is available [9]. Meanwhile, no high-quality studies comparing the AUS and adjustable male sling are available.

Therefore, the present study aimed to compare the outcomes of the AUS AMS800 and the third generation of the adjustable male sling ATOMS^TM^ at a tertiary center.

## 2. Materials and Methods

This was an observational retrospective cohort study with two arms. Retrospective data from clinical registers of male patients subjected to implantation of either an AUS AMS 800^TM^ or an adjustable male sling (ATOMS^TM^) in a single tertiary referral centre for the treatment of stress or mixed urinary incontinence (with prevalence of the stress component) were collected in a dedicated database. The prostheses which were used were the AMS 800^TM^ AUS and the ATOMS^TM^ adjustable male sling.

Inclusion criteria were: Patients affected by post-surgical genuine stress urinary incontinence (SUI) or mixed urinary incontinence (MUI) with prevalence of stress incontinence; symptoms refractory to pelvic floor training and conservative treatment; both naïve patients and patients with previous failure of anti-incontinence surgery.

Exclusion criteria were: Patients not subjected to preoperatory urodynamic exam; bladder compliance lower than 40 mL/cmH20 [10]; uncontrolled detrusor overactivity (DO); neurogenic bladder; <12 months total follow-up; incomplete data; patients subjected to surgery before year 2000.

Urinary incontinence was evaluated by the pad test/24 h and the pad count/24 h, and patients’ satisfaction was evaluated using the patient global impression of improvement questionnaire (PGI-I). Functional outcomes were evaluated at 12 months and at the last follow-up. PGI-I was evaluated at the last follow-up: PGI-I 1 or 2 was defined as “complete satisfaction” and PGI-I 3-4 was defined as “moderate satisfaction”, while PGI-I > 4 was considered “failure”. “Dry” outcome was defined as pad test/24 h < 10 mL. Complications were divided between “intraoperative” and “postoperative complications” and were categorized according to Clavien–Dindo classification. Prosthesis survival was defined as the time span between implantation and either the end of follow-up or the explantation. Follow-up was defined as the time span between implantation and either the explantation of the prosthesis or the end of clinical observation. The principal outcome was the difference in terms of postoperative complications, reintervention, explantation and survival between prostheses.

As secondary outcomes, we considered the differences in terms of functional outcomes (pad test/24 h, pad count/24, patients’ satisfaction) between prostheses.

### Statistical Analysis

Data were presented as means (SD) or medians (IQR) according to variable distribution (normal/non-normal). Variable distribution was assessed through multiple Shapiro–Wilk tests.

In order to enhance homogeneity between the two groups and to minimize selection bias, a propensity score to be subjected either to the AUS or the sling implantation was calculated. The covariates used for the creation of the propensity score were preoperatory pad test/24 h, previous pelvic radiotherapy, previous anti-incontinence surgery and age at surgery. Patients were matched according to their propensity scores with a maximum tolerance of 0.05. Comparative analysis for paired samples was then performed, using Chi-squared test for categorical variables and Wilcoxon test for paired samples or paired t-test for continuous variables.

Prosthesis survival at 60 months was assessed using the Kaplan–Meier estimator, and the difference between prosthesis survival was evaluated using the log-rank test.

Statistical analysis was performed using IBM SPSS statistical software version 25.0. *p* ≤ 0.05 was defined as statistical significance.

## 3. Results

The analysis of clinical registries allowed for the identification of 125 AUS implantations in 112 patients from 1995 to 2022 and 162 ATOMS^TM^ implantations in 162 patients from 2014 to 2022. After the application of the exclusion criteria, 94 AUS implantations in 81 patients and 95 ATOMS^TM^ in 95 patients were included in the study. Thereafter, propensity score matching was performed, and a cohort of 98 patients (49 for each intervention group) was obtained (flowchart is shown in Figure 1).

In Table 1, an overview of the baseline characteristics of the study populations before and after PSM is provided. The intervention which had caused urinary incontinence was open radical prostatectomy in most cases in both arms (a description of previous interventions is provided in Table 2). Concerning the surgical outcomes, the operative time was 56 ± 11 min in the ATOMS^TM^ group and 100 ± 19 min in the AUS group (*p* < 0.001). In the ATOMS^TM^ group, at the last follow-up, the mean number of refills was 2.75 ± 2.1, while the mean pressurization was 15.7 ± 10.3 mL. In the AUS group, 4 cm and 4.5 cm cuffs were the most used. A double bulbar cuff was used twice, while one single transcorporal implantation of the cuff was performed. Hospitalization was 2.6 ± 0.9 in the AUS group and 2.3 ± 1 in the ATOMS^TM^ group (*p* = 0.45). The mean follow-up was 43 ± 35 months.

### Safety, Complication, Reintervention, Survival

As shown in Table 1, the patients’ baseline characteristics were balanced after PSM. The preoperative pad test/24 h was 554 ± 297 mL in the AUS group and 522 ± 210 mL in the ATOMS^TM^ group (*p* = 0.37). PSM paired analysis showed that the postoperative pad test at the last follow-up was 100 ± 158 mL in the AUS group and 125 ± 156 mL in the ATOMS^TM^ group (*p* = 0.47). Urinary loss decrease/24 h was 320 ± 45 mL in the AUS group and 217 ± 31 mL in the ATOMS^TM^ group (*p* = 0.25). There were 22 (44.9%) dry patients in the AUS group and 11 (22.5%) in the ATOMS^TM^ group (*p* = 0.03). The patients declared themselves well improved (PGI-I 1-2) in 40 cases in the ATOMS^TM^ group (81%) and in 35 cases (71%) in the AUS group (*p* = 0,78). It is worth noting that there were no significant differences in continence between the patients with genuine SUI and MUI (in the case of well-controlled OAB). The functional outcomes are reported in Table 3.

An overview of the complications and reinterventions is reported in Table 3. The overall complications did not differ between the two groups (*p* = 0.09): there were 24 (49%) in the AUS and 17 (34.7%) in the ATOMS^TM^ group. On the contrary, when only considering the high-grade complications (Clavien–Dindo ≥ 3b), a statistically significant difference in favor of the ATOMS^TM^ group can be observed (*p* = 0.01). The high-grade complications in the AUS group consisted of urethral erosion, urethral atrophy and mechanical failure, and these were classified as Clavien 3b because they required the explantation or substitution of the prosthesis. In the ATOMS^TM^ group, the complications graded ≥ 3b consisted of refractory perineal pain, erosion and mechanical failure (deflating). Meanwhile, port repositioning using local anaesthesia due to displacement was classified as Clavien 3a, since this was performed in an outpatient setting using local anaesthesia. At the last follow-up, 10 slings (20.4%) and 24 AUSs (49%) required reintervention (*p* < 0.001). In the ATOMS^TM^ group, five reinterventions involved port repositioning, three involved prosthesis revision for inefficacy, and two involved explantation for refractory perineal pain. Prosthesis survival (until reintervention) at 60 months was estimated using the Kaplan–Meier function (Figure 2): this was 67 ± 7% in the ATOMS^TM^ group and 53 ± 6% in the AUS group, with there being no statistically significant difference between the two groups (*p* = 0.11). Survival until explantation (Figure 3) at 60 months was 82 ± 9% in the ATOMS^TM^ group and 67 ± 7% in the AUS group, with there being a statistically significant difference between the groups (*p* = 0.03).

## 4. Discussion

In this comparative study, we found that the outcomes of the two prostheses in relation to complications were similar when all grades were considered. However, the implantation of the AUS was linked to a higher occurrence of severe complications (36.7% vs. 14.3%) and a more frequent need for further interventions (49% vs. 20.4%). Notably, regarding the ATOMS^TM^ prosthesis, the rate of reinterventions was as low as 11.6% when port revisions were excluded. Furthermore, considering functional outcomes, our data suggest that opting for the ATOMS^TM^ device over the AUS may not greatly impact functional results. Our propensity-score-matched comparison between the two groups did not reveal any notable differences in terms of the postoperative 24 h pad tests (100 ± 158 mL in the AUS group and 125 ± 156 mL in the sling group) or the relative reduction in 24 h pad test volume (320 ± 45 mL in the AUS group and 217 ± 31 mL in the sling group).

Nonetheless, it is worth noting that a significantly higher proportion of “dry” patients were observed in the AUS group (44.9% in the AUS group compared to 22.5% in the sling group).

The heightened occurrence of reinterventions in the patients who underwent AUS procedures as opposed to those who underwent ATOMS^TM^ interventions found in this study accords with the results of a study conducted by Esquinas C. et al. This particular study involved 102 patients who underwent ATOMS implantation and 27 individuals who received AUS implants. The findings of the study revealed that the revision rate stood at 6.9% in the ATOMS^TM^ group, in contrast to a notably higher rate of 22% within the AUS group, with there being a follow-up of 34.9 ± 15.9 months. The prosthesis survival at 5 years was 81.7% in the ATOMS group and 69.9% in the AUS group, a finding similar to the results of our study [7]. Similarly, mounting evidence regarding the ATOMS^TM^ device, which can rely on large cohorts and for long follow-up periods, lends further credence to our data. A recent article published in 2023 by Giammò et al. [11] focused on a retrospective cohort of 99 patients subjected to ATOMS^TM^ implantation with a median follow-up of 62.9 months (47.5–75.9). This study recorded a 60-month survival rate of 87.9% and a 13.1% occurrence of late high-grade complications when port revisions were omitted. Similarly, a multicentric study published in 2020 by Angulo et al. [12] studied the outcomes of a large cohort of 155 patients subjected to ATOMS^TM^ implantation with a long follow-up of 60.6 ± 18.4. In the study, an 11.6% explantation rate and a 5-year survival rate of 86.3% (79.7–90.9) were recorded. Analogously, our data concerning the AUS are comparable to those of the available literature. An article published in 2017 by Suh et al. [13] found a 5-year prosthesis survival rate of 67% in a cohort of 155 patients subjected to AUS implantation.

In terms of continence outcomes, our data align quite closely with findings in the existing literature. Regarding the artificial urinary sphincter (AUS), the study by Suh et al. reported a continence rate of over 60%. However, it is worth noting that their definition of continence was more lenient compared to our study, as it did not require the use of pads. In other case series, continence rates have been reported to be as low as 20% [9]. Similarly, in the ATOMS^TM^ device studies by Giammò et al. and Angulo et al., continence rates ranging from 46% to 75% were found, depending on the definition of continence, results which outperformed those of our study. Nevertheless, the fact that we employed a stricter definition of “dry” in our study can account for this disparity.

Our finding that the implantation of the AUS is associated with a higher risk of urethral erosion and, conversely, our finding regarding the low number of urethral complications associated with the ATOMS^TM^ may represent interesting findings for guiding patients’ device selection and counseling. Indeed, patients with an increased risk of urethral injuries, such as patients who perform intermittent self-catheterization (ISC), are not suitable for the bulbar AUS; therefore, the ATOMS^TM^ can represent a valuable alternative. Indeed, the use of the ATOMS^TM^ device in patients with neurogenic bladders has already been carried out in a recent pilot study by Ammirati et al., which confirmed the safety of this device in this category of patients [14]. Indeed, port revision is the most frequently required reiterated procedure after ATOMS^TM^ implantation, and this does not systematically require the explantation of the prosthesis, since the mere displacement of the port without infection can be managed conservatively even with an increased risk of infection and re-extrusion [11].

A further important aspect to consider is the necessity of activating the AUS to initiate the bladder. Indeed, the AUS requires optimal patient compliance and proficient manual dexterity in order for it to be properly used. Therefore, the selection of a device that obviates the need for manual activation becomes particularly pertinent in the case of older or less physically capable patients, as well as those dealing with impaired cognitive function. In light of this, the availability of a device like the ATOMS^TM^ device, which can provide satisfactory results without requiring manual activation, is of particular relevance.

In this study, notwithstanding the several positive features of the ATOMS^TM^ device, the AUS still demonstrated better continence outcomes in terms of dryness. We believe that more patients achieved dryness in the AUS group compared to the ATOMS^TM^ group, despite similar postoperative pad-test results and reductions in urinary leakages, because the AUS AMS 800^TM^ cannot be adjusted once it is implanted. This means that a well-functioning AUS can provide excellent results but, if persistent urinary leakage occurs, the only way to rectify the situation is through a surgical revision. This characteristic serves as a significant limitation of the AUS AMS 800^TM^. Some recent adjustable devices such as the AUS Victo and Victo+ [15,16] have been introduced with the aim of addressing this limitation. Nevertheless, the current body of literature lacks high-quality prospective studies validating the efficacy and safety of these devices. On the contrary, the ATOMS^TM^ can be progressively adjusted and pressurized until satisfactory continence is achieved. However, patients should be aware of the possible need for the progressive pressurization of the device until continence is achieved, which may take several months.

Furthermore, although our study did not specifically aim to conduct a cost-effectiveness analysis, it is important to acknowledge that there is a noteworthy disparity in costs between the two prostheses. In our country, Italy, the prices of the respective devices are 4000 EUR for the ATOMS^TM^ device and 10,000 EUR for the AUS AMS 800^TM^. It is worth highlighting that the evaluation of costs extends beyond the initial prosthetic expenses. The potential for more frequent revisions and hospitalizations following AUS implantation must also be factored into a comprehensive cost assessment.

Ultimately, the findings derived from our study hold potential significance as a valuable informational resource for facilitating informed discussions concerning diverse interventions for postoperative male SUI) and MUI. According to our data, both prostheses can be very effective and can provide high satisfaction to patients (PGI-I 1 or 2 was obtained in 81% patients in the sling group and 71% in the AUS group), with both prostheses achieving comparable results in terms of the postoperative pad test/24 h. Nonetheless, the AUS demonstrated a higher potential for achieving complete continence than the ATOMS^TM^. This distinction should be understood alongside the AUS device’s associated increased risk of revision and explantation. Conversely, the ATOMS^TM^ is a valuable option in patients who require a less invasive procedure and in patients with an increased risk of urethral injury or limited manual dexterity, with the device able to maintain good functional outcomes.

We are aware that the present article is not devoid of limitations. First, the retrospective design of the study limits the level of evidence obtained. Moreover, the inhomogeneity of the periods during which the implantations were performed can represent a bias in our analysis. However, several precautions were adopted in order to reduce the risk of bias, such as our applying strict exclusion criteria and performing propensity score matching to reduce selection bias. In any case, more high-quality prospective studies are needed in order to better understand the differences between prostheses and to determine which patients are more suited to either ATOMS^TM^ or AUS implantation.

## 5. Conclusions

The ATOMS^TM^ and AUS AMS 800 can both provide high satisfaction and comparable urinary leakage reduction in patients with male postsurgical SUI and MUI. While the AUS may be characterized by a higher dry rate, its increased risk of high-grade complications and reintervention should be clearly explained to patients during counseling. On the contrary, the ATOMS^TM^ can be successfully proposed to patients who require a less invasive procedure that aims to maintain good functional outcomes.

## Figures and Tables

**Figure 1 jcm-12-05489-f001:**
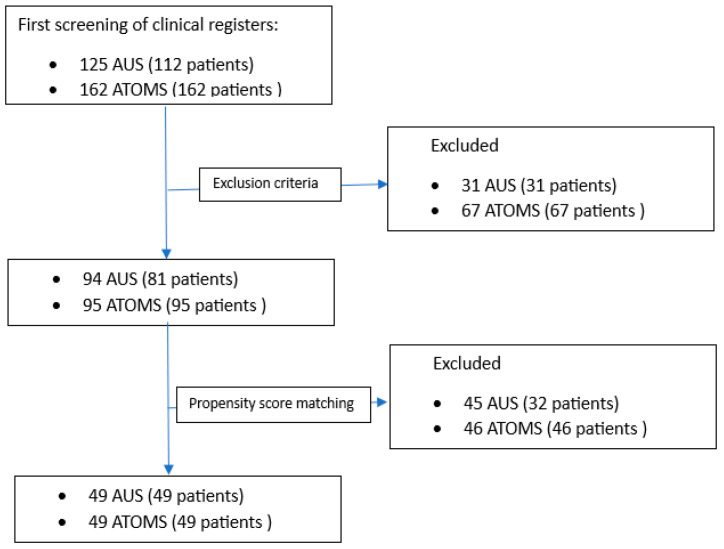
Inclusion criteria flowchart.

**Figure 2 jcm-12-05489-f002:**
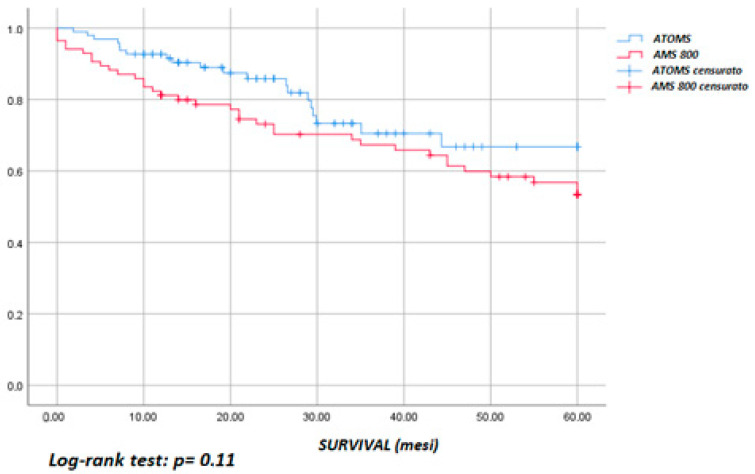
Survival until reintervention.

**Figure 3 jcm-12-05489-f003:**
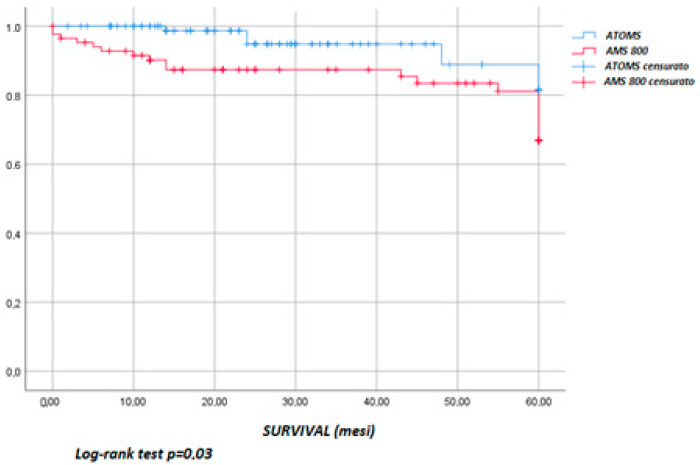
Survival until explantation.

**Table 1 jcm-12-05489-t001:** Baseline characteristics. AUS = artificial urinary sphincter; VLPP = Valsalva Leak Point Pressure; SSRIs = Selective Serotonin Reuptake Inhibitors; FD: first desire to void; MBC = maximum bladder capacity; OAB = overactive bladder.

Baseline				Propensity-Score-Matched Groups		
Baseline Characteristics	ATOMS Group	AUS Group	*p* Values	ATOMS Group	AUS Group	*p* Values
Age at surgery	71.5 ± 6.7	69 ± 10	*p* = 0.001	69.4 ± 7	69 ± 5.6	*p* > 0.05
Diabetes	4 (4.2%)	8 (9.8%)	*p* > 0.05	--	--	--
Pelvic radiotherapy	24 (25.2%)	31 (38.2%)	*p* = 0.02	15 (30.6%)	15 (30.6%)	*p* > 0.05
Hormone therapy	11 (11.6%)	28 (34.5%)	*p* < 0.001	--	--	--
OAB	20 (21%)	16 (19.7%)	*p* > 0.05	--	--	--
Detrusor overactivity	17 (17.9%)	14 (17.3%)	*p* > 0.05	--	--	--
VLPP (cmH20)	74.7 ± 44	53 ± 31.4	*p* < 0.001	--	--	--
FD (mL)	186 ± 21	175 ± 14	*p* > 0.05	--	--	--
MBC (mL)	255 ± 24	240 ± 55	*p* > 0.05	--	--	--
Pads/24 h (n)	4.5 ± 1.8	5.75 ± 3	*p* = 0.006	--	--	--
Pad test/24 h (g)	421 ± 196	646 ± 325	*p* < 0.001	522 ± 210	554 ± 297.7	*p* > 0.05
Antimuscarinics	15 (15.8%)	13 (16%)	*p* > 0.05	--	--	--
SSRIs	12 (12.6%)	3 (3.7%)	*p* > 0.05	--	--	--
Previous anti-incontinence surgery	58 (61%)	64 (79%)	*p* = 0.2	35 (71.4%)	31 (63.3%)	*p* > 0.05

**Table 2 jcm-12-05489-t002:** Previous surgeries. AUS = artificial urinary sphincter; RP = radical prostatectomy; LRP = laparoscopic radical prostatectomy; RARP = robot-assisted radical prostatectomy; TURP = transurethral resection of prostate; RC = radical cystectomy; BN = bladder neck; HIFU = high intensity focused ultrasounds.

Previous Surgery	ATOMS Group	AUS Group	Chi^2^
**Baseline Surgery**	*N* = 95 (100%)	*N* = 81 (100%)	
Open RP	68 (71.3%)	55 (67.9%)	
LRP	10 (10.5%)	6 (7.4%)	
RARP	6 (6.4%)	3 (3.7%)	
TURP	4 (4.3%)	8 (9.8%)	
RC	0	3 (3.7%)	
Simple prostatectomy	3 (3.2%)	5 (6.2%)	
Laser enucleation	4 (4.3%)	1 (1.3%)	***p* = 0.12**
**Anti-Incontinence Surgery**			
Pro-ACT	47(49.5%)	39 (48.1%)	
Fixed male sling	4 (4.2%)	5 (6.2%)	
AUS	6 (6.3%)	13 (16%)	
Urethral bulking	1 (1.05%)	7 (8.6%)	***p* = 0.26**
**Delayed Surgical Procedures**			
Uretrotomy	19 (20%)	21 (26%)	
BN incision	6 (6.3%)	10 (12.4%)	
HIFU	1 (1.05%)	2 (2.5%)	
TURP	2 (2.1%)	2 (2.5%)	

**Table 3 jcm-12-05489-t003:** Complications and functional outcomes. AUS = artificial urinary sphincter; PGI-I = patient global impression of improvement.

	ATOMS Group	AUS Group	*p*-Values
**Complications**			
Overall complication	17 (34.7%)	24 (49%)	*p* > 0.05
Clavien ≥ 3b	7 (14.3%)	18 (36.7%)	*p* = 0.01
Reintervention	10 (20.4%)	24 (49%)	*p* < 0.001
Explantation	2 (4%)	7 (14.2%)	*p* < 0.001
Months to reintervention	19.2 (7.2–29.4)	21 (6.5–52.5)	*p* > 0.05
**Functional Outcomes**			
Postoperative pad test/24 h (g)	125 ± 156	100 ± 158	*p* > 0.05
Pad test decrease/24 h (g)	217 ± 31	320 ± 45	*p* > 0.05
“Dry” outcome *n* (%)	11 (22.5%)	22 (44.9%)	*p* = 0.03
PGI-I 1–2 *n* (%)	40 (81%)	35 (71%)	*p* > 0.05

## Data Availability

Data are available on request due to privacy restrictions.

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
