# Peer review of "Comparison Study between Artificial Urinary Sphincter and Adjustable Male Sling: A Propensity-Score-Matched Analysis"

_jcm, 2023, doi:10.3390/jcm12175489_

Round 1

Reviewer 1 Report

Dear authors, congratulations for a very well designed and very appropriate comparative study between ATOMS and AUS. There are certain limitations for a comparative study like this but the authors have chosen a correct methodology to do so and also have presented the limitations of the study appropriately. Very nice nad important study.

On my opinion several comments must be taken into account, and would give rise to a second submission after minor corrections that could be interesting to improve the quality of therir presentation.

1. Reference 11 is well chosen as it deals with a very similar study, although this time the authors have performed propensity score matching that gives a much better scientific basis to their study. It is good however, that both studies go on the same line and with similar outcomes. However, speciual attention should be taken to correct a minor mistak on line 234 in the discussion section. The 22% revision rate was not in the ATOMS group, but in the AUS series in Esquinas et al.

2. There is sufficiently accumulated evidence to evaluate long term results of ATOMS as satisfactory, and some references shluld be added on that topic (PMID: 32496606 and 36983297). This topic is important, becasue ATOMS is a relatively recent therapeutic alternative for male stress incontinence after prostatectomy and obviously the body of evidence regarding AUS long-term outcomes is bigger, but still thogh ATOMS is doing also rather well on that. Even the durability of ATOMS could be superior to that of AUS. This cannot be proved, but is in consonance with the results of Reference 11 and this current article (Geretto et al).

3. Although it is not the scope of this study, economic data may also favour adjustable sling as compared to AUS. This could also be at least mentioned.

4. Also, the concept that ATOMS gives a broader perspective of use for patients with impaired cognitive function and also with diminished dexterity is another topic that needs be stressed. It is presented in the first paragraph of the discussion, but needs a better expanation.

5. This first paragraph in the discussion is excessively long (more than 45 lines) and mixes very different topics. It needs be rewritten more comprehensively, and this will also give a chance to improve English presentation.

6. An effort must be done to improve the layout (texts and tables) to follow MDPI format. Figures are correct.

In summary, the paper is novel and deals a very important issue. Also it should open a new window to call for prospective controlled analyses between the two devices compared.

See comment #5 

Author Response

The authors wish to thank Reviewer 1 for his insightful and useful review

  1. we noticed the mistake and corrected it in line 186
  2. We improved our disquisition about ATOMS outcomes by quoting the articles suggested by Reviewer 1 (lines 188-195 and 203-206)
  3. We mentioned the economic aspect which favours ATOMS over AUS (lines 236-241). Indeed costs are a very actual issue with whom we must deal daily and that often can influence clinical practice apart from clinical outcomes
  4. We highlighted this important aspect in lines 219-223
  5. As suggested, we rewrote the discussion clearer and more comprehensively and made an effort to iprove the introduction. Also, we reviewed our english presentation
  6. As suggested, we modified the document and tables more in compliance with MDPI guidelines

Reviewer 2 Report

Intro

48ff: The here described Atoms systems is the third generation of Atoms, this should be mentioned

65ff: The quotation of the Master trial does not fit into this context, not to mention the methodological issues of that study

M&M

76ff: was every degree of SUI included in the study? Also mild incontinence? Why?

88: the threshold of low bladder compliance is usually 20-25 cmH2

105ff: please explain why the complications are categorized as principal outcome and the treatment success (functional outcome) as secondary outcome

The presence of urodynamics was an inclusion criterion, but urodynamic data are completely lacking in the manuscript

Results

130: slings are commonly understood as fixed slings, e.g. AdVance, therefore Atoms should be named as adjustable sling. In this context it is not clear which types of sling are summarized table 2 as ‘Male sling’?

Table 2: it is surprising and unlucky that also cystectomy patients were included in the study what means a totally different condition and risk profile

158ff: as this is a retrospective study, was the indication for Atoms implantation at your institution not based on the degree of incontinence? The mean urine loss >500 ml seems to be quite high in this regard

204-207: probably a remainder from a past review

Discussion

224ff: the management of Atoms port complications is additional information that is neither the primary study subject, nor backed by the results. If at all it can be added at the end of the discussion.

234: Atoms means AUS here

239ff: the discussion should be better structured as a discussion of the own results first, in the context of the literature, and then a debate of the consequences, e.g. decision making, patient councelling, and alternatives. 

273ff: were these criteria already applied to the patients in the Atoms treatment group?

The effect of urge symptoms in MUI is not discussed at all, although MUI patients were included in the study, even the proportion of MUI and the characteristics are not mentioned.

Language not too bad but many mistakes both in grammar and word choice, revision by a native English speaker recommended

Author Response

The authors wish to thank Reviewer 2 for his time spent to comment our paper.

Introduction: as you suggested, we mentioned that the device which is currently in his third generation (line 50). However, we tried to edit the introduction into a more brief and fluent version with revised english, therefore we shortened the description of the device. We also cut the description of the MASTER RCT. We only mentioned it as it is the only RCT comparing AUS with a male sling

M&M: 

  • We included all consecutive patient subjected to AUS or ATOMS implantation with the exclusion criteria mentioned in the paper. In our ATOMS cohort, in fact, a few cases of mild incontinence were included. Nevertheless, after propensity score-matching those cases were excluded since any case of mild incontinence was recorded in the AUS group (therefore any matching was possible)
  • We agree with Reviewer 2 that the treshold for "low" compliance is generally 20ml/cmH20; however we included only patients with "normal" compliance which is generally recognized as >40ml/cmH20. We included the reference to the ICS Standard and we made it clearer in line 73-74
  • We decided to particularly focus on complications since one could argue that the difference  in terms of continence between AUS and ATOMS can be explained by the different indications of the two devices (Severe incontinence-->AUS; Moderate incontinence-->ATOMS). Therefore, the findings in terms of continence might be less relevant than the difference in terms of complications. Further directions of the analysis could lie in indagating the different outcomes in subgroups of patients 
  • In table 2 we specified more urodynamic parameter. As a matter of facts, the exclusion of patients with bladder compliance <40ml/cmH20 implies a previous urodynamic study 

 RESULTS:

  • We substituted the term "sling" with the term "ATOMS" in the whole paraghraph
  • We recognize that including RC patients introduces a bias in patients selection, nevertheless those patients were a few cases 
  • Indeed the mean 24h pad test is quite high. The first explanation is that as a tertiary referral center it is more likely for us to treat patients with severe incontinence and with recurrent urinary incontinence after surgery. Secondly, even if we are more likely to propose AUS to patients with severe SUI and ATOMS to patients with moderate SUI, we greatly value patient's preference. In this optic we noticed that patients very frequently choose ATOMS even if it could yeld to marginally inferior results than AUS. Therefore, particularly in the last years, we implant more and more ATOMS also to patients with severe SUI (with satisfaying results)
  • We corrected the mistake

DISCUSSION

  • We shortened and postponed the description of port complication management as suggested (lines 213-216)
  • We corrected the mistake (Line 186)
  • We edited the discussion paragraph as suggested, trying to make it more fluent and easy to read 
  • In our clinical practice we largely value patients couselling. We are used to discuss pros and cons of both the approaches and choose with the patient the most suitable for every case.
  • We added in table 2 the number of patients with OAB. We mentioned in our results that the presence of OAB did not influence outcomes if the urgency component is well controlled (which was one of our inclusion criteria)